# Immunotranscriptomic Profiling of *Spodoptera frugiperda* Challenged by Different Pathogenic Microorganisms

**DOI:** 10.3390/insects16040360

**Published:** 2025-03-31

**Authors:** Yan Tang, Qi Zou, Guojie Yu, Feng Liu, Yu Wu, Xueyan Zhao, Wensheng Wang, Xinchang Liu, Fei Hu, Zengxia Wang

**Affiliations:** 1College of Resource and Environment, Anhui Science and Technology University, Chuzhou 233100, China; ty20000314@163.com (Y.T.); yuguojie2001@163.com (G.Y.); 2College of Agriculture, Anhui Science and Technology University, Chuzhou 233100, China; zouqi1028@163.com (Q.Z.); lf030117@163.com (F.L.); 15551037678@163.com (Y.W.); 18726708153@163.com (X.Z.); 13030639220@163.com (W.W.); 19339171873@163.com (X.L.); 3Institute of Plant Protection and Agro-Products Safety, Anhui Academy of Agricultural Sciences, Hefei 230031, China; hufly0224@163.com; 4Anhui Engineering Research Center for Smart Crop Planting and Processin Technology, Anhui Science and Technology University, Chuzhou 233100, China

**Keywords:** *Spodoptera frugiperda*, pathogenic microorganisms, transcriptome, induced expression, immune genes

## Abstract

*Spodoptera frugiperda* is a highly destructive agricultural pest that poses a significant threat to global crop production. Current control strategies primarily rely on chemical pesticides, which can harm the environment and promote pest resistance. To explore safer and more effective alternatives, this study investigated the immune defense mechanisms of *S. frugiperda* under bacterial and fungal infections. Using advanced genetic analysis techniques, it was found that *S. frugiperda* activates a large number of immune-related genes upon infection, enabling it to recognize harmful microorganisms, transmit danger signals, and combat infections. The study also identified several genes potentially involved in key defense pathways and processes. These findings provide insights into the immune mechanisms of *S. frugiperda* and suggest potential genetic targets for developing eco-friendly pest control strategies such as enhancing the effectiveness of biopesticides by disrupting the pest’s immune system. This research offers new approaches to protecting crops while reducing the environmental and health impacts of chemical pesticides.

## 1. Introduction

*Spodoptera frugiperda*, commonly known as the fall armyworm, is a highly migratory agricultural pest of major significance, globally recognized by the Food and Agriculture Organization (FAO) as a severe threat. It is characterized by a broad host range, strong migratory ability, rapid spread, high reproductive rate, and significant resistance to pesticides, posing substantial risks to agricultural production in many countries [1]. Currently, chemical pesticides are the primary method for controlling *S. frugiperda*. However, excessive use of chemical pesticides has led to increased pest resistance and resurgence, disrupted the natural balance of ecosystems, and caused severe environmental pollution [2]. Biological control is an effective strategy for the sustainable prevention and control of *S. frugiperda*, and the application of biological agents such as microorganisms in particular shows a promising prospect [3,4]. Among the insecticides recommended by China’s Ministry of Agriculture and Rural Affairs for emergency control of *S. frugiperda*, six are biopesticides. These include five pathogenic microorganism-based products and one viral product: *Mamestra brassicae* nucleopolyhedrovirus, *Bacillus thuringiensis* (Bt), *Metarhizium anisopliae*, *Beauveria bassiana* (Bb), and *Empedobacter brevis*. Among these, *B. thuringiensis* is the most widely used and successful microbial insecticide, known for its high specificity, safety, and lack of residues [5].

Microorganisms can cause the death of *S. frugiperda*, but the pest simultaneously activates immune responses to resist and eliminate the invading pathogens [6,7]. Through long-term co-evolution, insects have developed a highly efficient and comprehensive innate immune system to defend against microbial and parasitic infections. Bai Yaoyu and colleagues discovered that injecting *Escherichia coli* (Ec) affects the cellular immune functions of *S. frugiperda* [8], while the invasion of *Steinernema carpocapsae*, a nematode species, leads to a “decrease–increase–decrease” trend in the hemolymph *phenoloxidase* activity of *S. frugiperda* larvae [9]. Pathogenic microbial invasion triggers a series of humoral and cellular immune responses in the host, stimulating the production of melanin and various antimicrobial peptides. The robust immune system of the host is one of the most critical factors limiting the “high virulence” of pathogens. Currently, research on the immune functions of *S. frugiperda* remains in its early stages. Therefore, an in-depth investigation into the composition and molecular regulatory mechanisms of the immune system of *S. frugiperda* is crucial for improving the efficacy of pathogenic microorganisms and developing novel, specific-target, and highly efficient biopesticides.

This study focuses on transcriptome sequencing of *S. frugiperda* infected by different pathogenic microorganisms, identifying immune-related genes through differential expression analysis and conducting bioinformatic analyses and expression validation of these genes. The aim is to uncover the molecular mechanisms underlying the interaction between pathogenic microorganisms and *S. frugiperda*, thereby providing new methods and theoretical foundations for utilizing pathogens in the biological control of *S. frugiperda*.

## 2. Materials and Methods

### 2.1. Test Insects

The *S. frugiperda* used in this experiment were collected from Fengyang County, Chuzhou City, Anhui Province (117.56° E, 32.86° N) and propagated for multiple generations indoors to establish an experimental population. The larvae of *S. frugiperda* were reared with artificial feed (150 g corn powder, 30 g yeast powder, 87 g soybean powder, 10 g sucrose and 15 g casein, 0.48 g cholesterol, 1.5 g ascorbic acid, 0.7 g choline chloride, 0.17 g inositol, 1 g sorbic acid, 1.4 g p-cyanobenzoic acid methyl ester, 0.5 g compound vitamin B, 0.25 g Weise’s salt, 2 mL rapeseed oil, 685 mL distilled water) [10] in an insectary at 27 ± 1 °C, relative humidity of 80% ± 10%, and a light–dark cycle of 14 L:10 D.

The bacteria used in this experiment were the Gram-positive bacteria *Staphylococcus aureus* (Sa, strain number: BNCC186335) and *Bacillus thuringiensis* (Bt, strain number: BNCC133158), the Gram-negative bacterium *Escherichia coli* (Ec, strain number: BNCC133264), as well as the fungus *Beauveria bassiana* (Bb, strain number: BNCC117565). The three bacterial strains were purchased from the Institute of Microbiology, Chinese Academy of Sciences. Before the experiment, all experimental equipment was sterilized by moist heat at a high temperature (121 °C for 20 min). The bacteria were transferred into a 250 mL conical flask containing LB liquid medium using a sterile pipette tip. Then, the flask was placed in a shaker at 37 °C and rotated at 200 rpm overnight for 12 h, and then the cultures were collected. The collected cultures were centrifuged at 8000 g for 5 min. Then, the supernatant was discarded. The pellet was inactivated by heating at 85 °C for 1 h. After that, the inactivated bacterial suspension was diluted with sterile PBS (pH 6.4, containing 7.7 mmol/L Na_2_HPO_4_, 2.65 mmol/L NaH_2_PO_4_, and 150 mmol/L NaCl) to a bacterial colony count of 3 × 10^6^ cells/mL for later use. *B. bassiana* was a gift from Teacher Hu Fei of the Institute of Plant Protection and Agricultural Product Quality and Safety, Anhui Academy of Agricultural Sciences. *B. bassiana* was placed in PDA medium and cultured at 27 °C. After collection in an enzyme-free EP tube, sterilized water (containing 0.5% Tween-80) was added, shaken thoroughly, filtered through sterilized cotton, and the filtrate was heated at 85 °C for 1 h and inactivated. Then, it was counted using a hemocytometer and diluted to 3 × 10^6^ cells/mL for later use.

### 2.2. Injection Experiment

The injection experiment of *S. frugiperda* larvae was conducted following the method of Sun and Bai (2020) [11]. Healthy fourth-instar *S. frugiperda* larvae of consistent size were randomly selected and divided into a PBS control injection group, a *S. aureus* injection group, an *E. coli* injection group, a *B. thuringiensis* injection group and a *B. bassiana* injection group. There were 5 larvae in each group treatment, and 3 biological replicates were set for each treatment. After disinfecting the abdominal feet with 70% alcohol, the *S. frugiperda* larvae were placed on ice for 5 min for cryoanesthesia. We used a disposable sterile syringe with a size of 0.45 mm × 16 mm to draw 5 µL of inactivated bacterial suspension (approximately 3.0 × 10^6^ cells/mL). Then, it was injected into the abdominal proleg of the larvae through a micro-applicator (The Hand Microapplicator, Model PDE0003, is manufactured by Burkard Manufacturing Co., Ltd. in Rickmansworth, UK). Meanwhile, an equal volume of phosphate-buffered saline (PBS) injection was used as the control.

After injection, the wound was surface sterilized with 70% ethanol. Then, the injected larvae were placed separately in plastic tubes. They were provided with artificial feed and kept at 27 ± 1 °C, relative humidity of 80% ± 10% and a light–dark cycle of 14 L:10 D for 24 h. Then, the larvae were separately collected in centrifuge tubes and quickly frozen in liquid nitrogen. Subsequently, they were transferred to a −80 °C freezer for storage and future use.

### 2.3. Total RNA Extraction and Sequencing

Total RNA of *S. frugiperda* was extracted using the Trizol method. During the RNA extraction process, the ribosomal RNA (rRNA) depletion technique was employed. The quality and integrity of the total RNA were detected by 1% agarose gel electrophoresis. The concentration and purity of the total RNA were detected using Nanodrop 2000 (Thermo Fisher Scientific Inc. from Waltham, MA, USA), and the integrity of the RNA was accurately detected using Agilent 2000 bioanalyzer (Agilent Technologies, Inc. from Santa Clara, CA, USA). RNA samples with an RIN value between 7 and 9, an A260/A280 ratio of 1.8–2.0, and an A260/A230 ratio of ≥ 2.0 were considered as qualified samples and were subjected to transcriptome sequencing using the Illumina NovaSeq 2000 (Illumina, San Diego, CA, USA) platform.

### 2.4. Gene Expression Quantification and Differential Analysis

To ensure the quality and reliability of data analysis, according to the established quality control criteria, reads with adapters, reads containing N (where N represents undetermined base information), and low-quality reads (reads in which the number of bases with Qphred ≤ 5 accounts for more than 50% of the entire read length) were removed. After preprocessing, high-quality clean reads were generated. Subsequently, the clean reads from different samples were combined and subjected to de novo assembly analysis using the Trinity (Trinity v2.5.1) software to construct transcript sequences. The assembly quality of Trinity.fasta, unigene.fasta, and cluster.fasta was assessed using BUSCO software v5.8.2, and the accuracy and completeness of the assembly were evaluated based on the GC content and unigene sequence integrity. Gene expression levels were quantified by FPKM (Fragments Per Kilobase of transcript per Million fragments mapped). Differential gene expression analysis was performed using the DESeq2 software 1.46.0. The criteria for screening differentially expressed genes were FDR less than 0.05 and fold change (FC) greater than 1. The clusterProfiler 4.14.4 software was used to perform GO functional enrichment analysis and KEGG pathway enrichment analysis on the differentially expressed gene set, and a threshold of padj less than 0.05 was set for significant enrichment. For immune genes, a two-way clustering heatmap analysis was carried out with FDR less than 0.05 and fold change (FC) greater than 2.

### 2.5. Identification and Analysis of Immune-Related Genes

In this study, all gene sequence information was sourced from this paper. For the whole-genome sequencing and gene annotation of *S. frugiperda*, the methods of Gouin et al. [12] were referred to. The Immunome Knowledge Base (https://ngdc.cncb.ac.cn/iaa/home accessed on 21 October 2024), an immune-related database, was utilized to compare and reference the genes to be identified with the information in these databases, providing a basis for the identification of the immune-relatedness of genes. CD-search (http://www.ncbi.nlm.nih.gov/Structure/bwrpsb/bwrpsb.cgi (accessed on 21 May 2024)) and SMART (http://smart.embl-heidelberg.de/ accessed on 6 May 2024) were used to examine conserved domains in candidate genes of non-redundant protein families. Signal peptides and transmembrane domains of different protein families were verified using SignalP 4.1 (http://www.cbs.dtu.dk/services/SignalP accessed on 15 June 2024) and TMHMM (http://www.cbs.dtu.dk/services/TMHMM/ accessed on 16 June 2024). The Fragments Per Kilobase of exon per Million reads mapped (FPKM) method was used to evaluate the gene expression in the sequenced *S. frugiperda* samples. The DESeq software was employed to analyze the differentially expressed genes among the samples. Heatmaps of the FPKM values of differentially expressed genes across treatments were generated using TBtools-II v2.136, and MEGA 11.0 was used to align homologous immune proteins of *S. frugiperda* with those of other insects. A phylogenetic tree of *S. frugiperda* immune family proteins and homologous proteins from other insects was constructed using the maximum likelihood method, and FigTree1.4.4 was used for visualization to clarify the evolutionary relationship of immune proteins between *S. frugiperda* and other insects.

### 2.6. Relative Quantification of Gene Expression by RT-qPCR

To verify the expression changes in the PGRP family genes after *S. frugiperda* is infected by microorganisms, fourth-instar *S. frugiperda* larvae were injected with *B. thuringiensis*, *B. bassiana*, *S. aureus*, and *E. coli*, and PBS was injected as a control. Thirty larvae were injected for each treatment, and three biological replicates were performed. After injection at 2 h, 4 h, 8 h, 12 h, 24 h, and 48 h, five larvae were collected, respectively, for dissection. After being placed on ice for 5 min for cryoanesthesia, the prolegs of the larvae were disinfected with 70% alcohol. The sterilized insect needles were used to puncture the abdominal feet of the larvae, and the prolegs were gently squeezed. The extruded hemolymph was dripped onto parafilm film, and then collected in a 1.5 mL centrifuge tube. Then, the body wall was dissected, and the midgut, fat body, and body wall were collected. The hemolymph was centrifuged at 12,000 g at 4 °C for 30 min, and the supernatant was taken.

Primers were designed using the NCBI online tool (https://www.ncbi.nlm.nih.gov/tools/primer-blast/ accessed on 11 July 2024). High-quality RNA samples were first processed using the same RNA extraction method as described above. Then, the TransScript One-step gDNA Removal and cDNA Synthesis SuperMix kit was used to reverse transcribe these RNA samples into cDNA. Subsequently, fluorescence quantitative PCR reactions were performed on the ViiA™ 7 fluorescence quantitative PCR instrument (From Applied Biosystems, a company under Thermo Fisher Scientific, located in Waltham, MA, USA) following the instructions of TB Green Premix Ex Taq II (Tli RNase H Plus) (From TaKaRa Bioengineering (Dalian) Co., Ltd., located in Ōtsu City, Shiga Prefecture, Japan). Three biological replicates and three technical replicates were set up for each reaction. The PCR reaction system (20 μL) consisted of 2 μL of cDNA, 10 μL of TB Green^®^ Premix Ex Taq™ II (2×), 0.8 μL each of forward and reverse primers, 0.4 μL of ROX Reference Dye I (50×), and 6 μL of sterile water. The reaction procedure was carried out using a two-step method: pre-denaturation at 95 °C for 30 s; denaturation at 95 °C for 3 s; annealing at 60 °C for 30 s and 40 cycles; and fluorescence signals were collected. Melting curve: fluorescence signals were collected every 6 s from 60 °C to 95 °C to draw the melting curve, and the specificity of the primers was judged based on the melting curve. In addition, the cDNA template was serially diluted by a factor of 10, and a standard curve for each pair of primers was drawn based on the fluorescence quantitative results. The amplification efficiency was calculated using the formula E = (10[−1/slope]−1) × 100. Finally, primers of candidate genes were selected when the amplification efficiency was between 90% and 110% and the melting curve had only a single peak. RPL3 and RPL18 [13,14] were used as internal reference genes, and the sequences of gene primers are shown in Table 1.

Quantitative polymerase chain reaction (qPCR) was used to detect the expression levels of the peptidoglycan recognition protein (PGRP) family genes in different tissues of *S. frugiperda* after it was induced by microorganisms at different times. The relative expression levels were calculated using the 2^−ΔΔCt^ method [15].

### 2.7. Statistical Analysis

The differences between different time points or treatments were analyzed by two-way analysis of variance on the data using Prism 10 CN software. In our two-way analysis of variance, the independent variable was different microbial treatments, and the dependent variable was different time points. The Tukey test was used for post-hoc analysis, the Levene test was used to assess the homogeneity of variances, and the Shapiro–Wilk test was used to evaluate the normality of the data. Meeting the results of the above-mentioned tests simultaneously indicated that the assumptions were satisfied, and graphs were drawn.

## 3. Results

### 3.1. Transcriptome Sequencing Data Statistics and Analysis

The transcriptome data of the fourth-instar *S. frugiperda* larvae after injection were analyzed. The detailed data have been uploaded to the National Center for Biotechnology Information (NCBI) (https://dataview.ncbi.nlm.nih.gov/object/PRJNA1212310?reviewer=s3du3flre0ku0mdoquhhi164jm Uploaded on 31 October 2024). After filtering the raw data, 50,261,762; 46,235,856; 45,953,424; 49,723,532; and 44,794,776 high-quality reads were obtained in PBS, Ec, Bt, Sa, and Bb, respectively. In all treatments, the clean bases reached more than 6.72 Gb, the Q30 was above 95.72%, the Q20 was above 98.462%, the GC content was between 45.15% and 46.7%, and the error rate of each treatment was 0.01%. The percentage of reads mapped to the whole genome was between 78.16% and 82.51%. The percentage of reads mapped to the unique position of the reference genome (reads used for subsequent quantitative data analysis) was 75.23% in PBS, 77.81% in Ec, 76.04% in Bt, 75.05% in Sa, and 73.95% in Bb (Table 2 and Table 3). This indicates that the integrity of the assembly result is good and it can be used for further analysis.

### 3.2. Differentially Expressed Gene Analysis

A total of 10,453 differentially expressed genes were identified after treating *S. frugiperda* with pathogenic microorganisms. Comparative analysis of DEGs with the control group revealed that the Bb-treated group exhibited significantly more DEGs than the other treatment groups, with a total of 3593 DEGs. Among these, 2044 genes were upregulated, and 1549 genes were downregulated.

In the Sa-treated group, 2419 DEGs were identified, including 1734 upregulated genes and 685 downregulated genes. The Bt-treated group showed 2571 DEGs, with 1474 genes upregulated and 1097 genes downregulated. Lastly, the Ec-treated group had 1870 DEGs, with 994 genes upregulated and 876 genes downregulated (Figure 1).

### 3.3. GO Functional Enrichment Analysis

Gene Ontology (GO) describes the characteristics of gene products from three aspects: molecular function, cellular component, and biological process. These aspects are closely related to the distribution of differentially expressed genes in GO enrichment analysis. At the molecular function level, genes can be involved in functions related to immune recognition and the regulation of enzyme activity in the immune system. At the cellular component level, GO annotations describe the localization of gene products in specific substructures of immune cells. When immune cells interact, the cellular components related to the “immune synapse” are involved. At the biological process level, genes are involved in the activation process of immune cells and the inflammatory response process. The GO enrichment bar chart can show the distribution of differentially expressed genes enriched in GO terms. The hypergeometric test was used to conduct GO functional enrichment analysis of the differentially expressed genes induced by the infection of the fourth-instar *S. frugiperda* larvae with pathogenic microorganisms. It was found that the differential genes were mainly distributed in biological process (BP), cellular component (CC), and molecular function (MF) (Figure 2). In the Bb treatment, 707 unigenes were annotated to biological process, 125 to cellular component, and 1198 to molecular function. The main distribution subcategories were the oxidation–reduction process in BP, the extracellular region in CC, and the oxidoreductase activity in MF. In the Bt treatment, 477 unigenes were annotated to biological process, 88 to cellular component, and 686 to molecular function. The main distribution in MF was peptidase activity. In the Ec treatment, 481 unigenes were annotated to biological process, 78 to cellular component, and 712 to molecular function. The main distribution in BP was proteolysis. In the Sa treatment, 368 unigenes were annotated to biological process, 97 to cellular component, and 617 to molecular function. The main distribution in BP was transmembrane transport. After different pathogenic microorganisms infected the fourth-instar *S. frugiperda* larvae, the change levels of differentially expressed genes varied. However, most of them were concentrated in the molecular function process, while relatively fewer differentially expressed genes were found in the biological process and cellular component.

### 3.4. KEGG Pathway Enrichment Analysis

In the KEGG pathways, the Toll-like receptor signaling pathway is a crucial component of innate immunity. Members of the NOD-like receptor (NLR) family are capable of recognizing pathogen-associated molecules within cells. Collaborating with the Toll-like receptor signaling pathway, they jointly form the sensing and response network of innate immunity to pathogens, rapidly initiating the defense mechanism in the early stage of the immune response. The T-cell receptor signaling pathway is the core pathway for T-cell activation in adaptive immunity. The B-cell receptor (BCR) signaling pathway plays a key role in the humoral immunity of adaptive immunity. The hypergeometric distribution test was used for the DEG (differential gene) results obtained after the induction of *S. frugiperda* by microorganisms (*p* < 0.05). The 20 most significant KEGG pathways were selected to draw a scatter plot and conduct enrichment analysis. The results showed that for the 3593 DEGs in the Bb treatment group, the KEGG pathways with the largest numbers of differentially expressed genes were *Carbon metabolism* (67), *Drug metabolism-other enzymes* (47), and *Biosynthesis of amino acids* (43). Meanwhile, the largest number of differentially expressed genes were enriched in these pathways. Among the 2571 DEGs in the Bt treatment group, the significantly enriched KEGG pathways were *Carbon metabolism* (45), *Biosynthesis of cofactors* (39), *Neuroactive ligand-receptor interaction* (37), etc. Among the 2419 DEGs in the Sa treatment group, the genes were enriched in pathways such as *Carbon metabolism* (37), *Motor proteins* (34), and *Oxidative phosphorylation* (24). Among the 1870 DEGs in the Ec treatment group, the significantly enriched pathways were *Neuroactive ligand–receptor interaction* (36), *Lysosome* (36), *Peroxisome* (26), etc. The results of this study showed that the number of genes expressed in the carbon metabolism pathway was the largest in all treatments. Some intermediate products and enzymes in the carbon metabolism process can act as signaling molecules or regulatory factors and participate in the regulation of immune signaling pathways [16] (Figure 3).

### 3.5. Screening of Immune-Related Genes

By comparing the immune-related gene sequences of known model insects, 598 immune-related genes of various categories were identified from the transcriptome sequences of *S. frugiperda* (see Table A1). Based on their functions, these genes were categorized into four major groups: pattern recognition receptors, immune effectors, signal transduction factors, and immune regulatory factors. Signal transduction factors include components of the IMD, Toll, JAK/STAT, JNK, RNA interference, and autophagy immune pathways.

Specifically, the study identified 98 pattern recognition receptors belonging to 12 gene families, accounting for 16.39% of the total immune-related genes. The *serine protease inhibitor* family with clip domains and the *serine protease* family were found to have 20 and 30 members, respectively, which regulate the amplification and attenuation of extracellular immune signals. These immune regulatory factors account for 24.08% of the total immune-related genes.

In the IMD, Toll, JAK/STAT, JNK, RNA interference, and autophagy immune signal transduction pathways of *S. frugiperda*, 92 components responsible for signal transduction were identified, representing 15.38% of the total. Immune effectors in insects, such as *antimicrobial peptides*, *lysozymes*, *melanin*, and *antioxidative molecules*, were also identified. A total of 264 immune effectors, accounting for 44.15% of the total immune-related genes, were identified in this study.

As shown in Figure 4, when *S. frugiperda* is infected by pathogenic microorganisms, its pattern recognition receptors, immune effectors, signal transduction factors, and immune regulatory factors are all activated. Among these, genes related to immune effectors constitute the largest proportion, indicating that through long-term biological evolution, *S. frugiperda* has developed a relatively complete innate immune system.

### 3.6. Analysis of Immune-Related Genes

The immune gene families of *S. frugiperda* are complex, with functional differences observed among genes within the same family. This study conducted phylogenetic and expression variation analyses of immune genes in comparison with known model species to better understand the functions and expression trends of immune-related genes in *S. frugiperda*.

#### 3.6.1. Pattern Recognition Receptors

Insects rely on unique *pattern-recognition receptors (PRRs)* to detect *pathogen-associated molecular patterns (PAMPs)* on the surfaces of microorganisms. In the transcriptome of *S. frugiperda*, 12 types of PRRs were identified (see Table A1). Among them, *Integrin* was the most abundant, with 27 unigenes, followed by *ubiquitin-conjugating enzyme* (20), *DSCAM* (11), *PGRP* (9), *SR* (9), *Galectin* (8), *Vitellogenin* (5), *βGRP* (3), *C-type lectin* (2), *ApoLp* (2), TEP (1), and *Croquemort* (1). The transcriptional levels of pattern recognition receptor (PRR) genes vary under different pathogenic microorganism infections (Figure 5). In the Bb treatment, genes such as *PGRP-LB1, LB, S1, S2; SR-2; Integrin-1, 2, 4, 5, 6; Galectin-1, 2; DSCAM-3; Vitellogenin-2; βGRP-1; CTL*; and *ubiquitin-conjugating enzyme* were significantly upregulated, while *Integrin-3, Vitellogenin-1,* and *βGRP-2* were significantly downregulated. In the Bt treatment, *PGRP-LB1, S2, SR-2, 3, 4; Integrin-2; Integrin-7; Galectin-2; DSCAM-1, 3; Vitellogenin-2; CTL*; and *ubiquitin-conjugating enzyme* were significantly upregulated, while *SR-1* and *βGRP-2* were significantly downregulated. In the Ec treatment, *PGRP-LB1, S1, S2, SR-2; Integrin-1, 7; DSCAM-1, 3; Vitellogenin-2*; and *ubiquitin-conjugating enzyme* were significantly upregulated, and *βGRP-2* was significantly downregulated. In the Sa treatment, *PGRP-LB1, S1, S2; Integrin-4; DSCAM-1, 2, 3*; and *ubiquitin-conjugating enzyme* were significantly upregulated, and *SR-1* was significantly downregulated. The results indicate that different recognition receptor genes may participate in different immune pathways during the immune response to recognize pathogenic microorganisms.

The most notable feature of the *PGRPs* family is the presence of a T4 bacteriophage lysozyme domain [17]. In insects, *PGRPs* are classified into three types based on molecular weight: short (S), intermediate (I), and long (L) [18].

The short type (S), with a molecular weight of approximately 20–25 kD, typically contains signal peptides, lacks transmembrane domains, and functions as small, secreted extracellular proteins. The intermediate type (I) has a molecular weight of approximately 40–45 kD. The long type (L), generally exceeding 90 kD in molecular weight, can be further subdivided into two subtypes: intracellular proteins lacking both signal peptides and transmembrane domains, and transmembrane proteins containing signal peptides and transmembrane domains, which exist as transmembrane proteins [19].

In the transcriptome unigene data of *S. frugiperda*, nine *PGRP* family genes were identified and named *PGRP-L1*, *L2*, *LB*, *LB1*, *LB2*, *LE2*, *S1*, *S2*, and *S3* based on sequence characteristics and multiple sequence alignments. Phylogenetic tree analysis reveals that the *PGRP* gene family mainly clusters with Lepidoptera insects (Figure 6a). *PGRP-S3* of *S. frugiperda* has a relatively close genetic relationship with *BmPGRP-L1*, *PGRP-S2* with *BmPGRP-S2*, and *DmPGRP-LC* with *PGRP-L1*. The *BmPGRP-L1* gene is involved in the immune response of *Bombyx mori* to the Gram-negative bacterium *E. coli*, participating in the IMD signaling pathway in the body wall and head [20]. *BmPGRP-S2* can regulate the expression of antimicrobial peptides (AMPs) in the body wall of *B. mori* and is involved in the activation of the IMD signal transduction pathway in the body wall of *B. mori* [21], while *DmPGRP-LC* activates the IMD pathway through membrane-bound or intracellular receptors [22], suggesting that the corresponding genes of *S. frugiperda* may have similar functions. As shown in Figure 6b, the amino acid sequences of *PGRP-S1*, *S2*, and *S3* genes contain signal peptides, suggesting that they may be secretory extracellular proteins of the short type. This indicates that they might be secreted outside the cell to participate in the melanization reaction after receiving signals. *PGRP-LB2* contains a signal peptide and a transmembrane domain, indicating that it may exist in the form of a transmembrane protein to activate the immune signaling pathway.

*βGRP*, also known as *Gram-negative binding protein* (*GNBP*), contains two main conserved domains: an N-terminal *β-1,3-glucan recognition* domain and a C-terminal *β-1,3-glucan recognition* domain lacking catalytic residues. These domains are responsible for recognizing the cell wall polysaccharides of Gram-negative bacteria or β-1,3-glucans in fungi [23,24].

In the transcriptome data of *S*. *frugiperda*, three homologous *βGRP* genes were identified. Phylogenetic analysis showed that *βGRP1* of *S*. *frugiperda* clustered together with *HamGRP-2* of *H*. *armigera*, while *βGRP2* clustered together with *BmorGRP-1* and *BmorGRP-2* of *B*. *mori*. Previous studies have demonstrated that *BmorGRP-1* of *B*. *mori* can bind to fungi and bacteria, activating the *phenoloxidase*-mediated melanization reaction [25]. Therefore, *βGRP2* is likely to be a key recognition receptor in the immune signaling pathways of pathogenic microorganisms. Meanwhile, *βGRP3* clustered together with the *βGRP* family of Aedes aegypti, indicating their evolutionary similarity (Figure 7a). The sequence structure diagram shows that the N-terminal ends of all three genes contain signal peptide sequences, which indicates that the proteins expressed by these genes can be secreted into the hemolymph to perform pathogen recognition functions (Figure 7b).

#### 3.6.2. Immune Regulatory Factors

Regulatory factors in insect plasma include *serine proteases*, their non-catalytic homologs (*serine protease homologs*, *SPHs*), and *serine protease inhibitors* (*serpins*). *SPs*, one of the largest protein families in insects, amplify invading immune signals through proteolytic cascade reactions, especially those containing clip domains [26]. *Serpins*, as inhibitors of *SPs*, attenuate immune signals and provide feedback regulation. Members of a protein superfamily, *serpins* often form covalent complexes with *SPs*, blocking *SP* cascades to precisely regulate the *prophenoloxidase* cascade and the Toll pathway [27].

Thirty *SP* genes, five *Elastase* genes, ninety-seven *Trypsin* genes, and twenty *serine protease inhibitor serpin* genes were screened from the transcriptome of *S. frugiperda* (see Table A1). As shown in the expression levels in Figure 8, the immune regulatory factors of *S. frugiperda* showed different transcriptional levels after being infected by different pathogenic microorganisms. In the Bb treatment, 19 genes were upregulated, with *Serpin-6* being significantly upregulated; 21 genes were downregulated, with the most significant downregulation occurring in *SP-5*. In the Bt treatment, 16 genes were involved in positive regulation, and 24 genes were involved in negative regulation. In the Ec treatment, 12 genes were downregulated and 11 genes were upregulated. In both the Bt and Ec treatments, the most significant downregulation occurred in *Trypsin-23*, and the significant upregulation occurred in *Serpin-6*. In the Sa treatment, 9 genes were downregulated and 8 genes were upregulated.

The results of the phylogenetic tree analysis show that the *SPs* of *S. frugiperda* are mainly clustered with, and have the closest genetic relationship to, those of lepidopteran insects such as *B. mori*, *Ostrinia furnacalis*, and *Helicoverpa armigera* (Figure 9).

Among them, *SfurSP-5* and *HarmSP-6* cluster into one branch. Xiong et al. found that *cSP6* in *H. armigera* is crucial for activating *prophenoloxidase* [27]. Thus, it can be inferred that *SfurSP-5* may also have a similar function and be involved in the activation of the melanization reaction. In the phylogenetic tree, *SfurSP-10* has the closest genetic relationship with *OfurSP-1*. As reported by CHU et al. [28], two *serine proteases* (*SP1* and *SP13*) mediate the melanization reaction of *O. furnacalis* in response to fungal invasion, indicating that *SfurSP-10* may be involved in the melanization reaction.

The *Serpin* genes of *S. frugiperda* are closely related to those of both lepidopteran and hymenopteran insects. Previous studies have shown that *MsSerpin-6* in *Manduca sexta* can block the upstream signal transmission of the *prophenoloxidase* cascade [29]. Heterologous transfer of *DmelSerpin77Ba* in *Drosophila* can lead to local melanization of the respiratory trachea and activation of the systemic Toll signaling pathway [30]. In the phylogenetic tree, *Sfurserpin-3* clusters with *DmelSerpin77Ba* and *Msserpin-6* (Figure 10a). It is inferred that *Sfurserpin-3* has similar functions and is involved in the fine-tuning of the melanization reaction and the Toll signaling pathway in this pest. Currently, in *Tenebrio molitor*, *TmolSerpin55* has been found to form a complex with the *serine protease SAE* in the *serine protease* cascade of the Toll pathway, jointly negatively regulating the formation of melanin regulated by the Toll signaling pathway. In *Drosophila*, the *Serpin-1* protein acts on the pattern recognition receptor *GNBP3*, which recognizes and binds to *β-1,3-glucan* on the fungal cell wall to activate the Toll pathway [31,32]. *SfurSerpin-1* clusters with these, from which it is inferred that it may be involved in the regulation of the Toll pathway in *S. frugiperda*. Among them, seven SPN genes have complete domains and signal peptide sequences, indicating that they function extracellularly (Figure 10b).

#### 3.6.3. Signal Transduction Factors

The innate immune signaling pathways in the model insect *Drosophila melanogaster* include four primary pathways: Toll, Imd, JAK/STAT, and JNK4. Among these, the Toll and Imd pathways have been gradually confirmed in many lepidopteran insects [33,34]. The Toll signaling pathway in insects is evolutionarily highly conserved and primarily defends against fungi and Gram-positive bacteria. In *S. frugiperda*, the Toll signaling pathway is relatively complete, encompassing genes encoding the extracellular cytokine *Spätzle*, transmembrane receptors such as *Toll proteins*, *tolloid-like proteins*, and *Toll-like receptors*, as well as intracellular signaling components *Tube*, *myeloid differentiation factor 88*, *Pelle kinase*, the *inhibitor molecule Cactus*, *Cactin*, *Pellino*, and the *NF-κB transcription factor dorSA1.*

The IMD pathway is a critical pathway for defending against Gram-negative bacteria. In *S. frugiperda*, key regulatory genes identified in this pathway include 1 *TAK1*, 2 *IKKs*, 1 *Sickie*, 1 *Akirin*, and 6 *Cullins* (see Table A1).

The JNK and JAK/STAT signaling pathways are also essential immune defense mechanisms in insects. In the transcriptome data of *S. frugiperda*, the core genes of the JAK/STAT pathway were identified, including those encoding JAK kinase (*Hopscotch*) and STAT factors, as well as the negative regulatory genes *SOCS* and *PIAS*.

In terms of expression levels, among the signal transduction factors, in the Bb treatment, the genes *Spätzle, Pelle, Activator Protein (AP)-1*, and *AP-2* were significantly upregulated, while *Toll-3* was significantly downregulated. After the Bt treatment, the gene *Toll-3* was downregulated. In the Ec treatment, *Spätzle* and *AP-1* were significantly upregulated. *Spätzle* showed an upward trend in the Sa treatment, and the gene *Toll-2* was the most significantly downregulated. Overall, the four innate immune signaling pathways coordinate with each other, are responsible for transducing danger signals, and ultimately stimulate the production of immune effector molecules (Figure 11).

Signal transduction factors can transmit the signals of external pathogen invasion into the cells to initiate an immune response. A total of 12 *Toll* genes have been identified in *S*. *frugiperda*. The results of the phylogenetic tree show that they are mainly closely related to Diptera and Lepidoptera (Figure 12a). Previous research results have shown that *PxylToll-6* of *P*. *xylostella* and *AgamToll-9* of Anopheles gambiae may be involved in the innate immune response against invading pathogens. The clustering results indicate that *SfurToll-9* of *S*. *frugiperda* may be involved in the innate immune response against invading pathogens.

*Spätzle* is involved in the functions of recognizing pathogens and regulating the intensity of the immune response in insect immunity. In *S. frugiperda*, we have identified 5 *spz* genes, which mainly cluster together with Lepidoptera insects (Figure 12b). *BmorSpz4* of *B. mori* can activate the intracellular Toll signaling pathway, and it is likely that *SfruSpz2* also has a similar function, influencing the host’s participation in the immune response.

#### 3.6.4. Immune Effector Factors

Microbial induction can lead to the production of numerous effectors, which are small-molecular-weight proteins. In the transcriptome of *S. frugiperda*, a total of 264 effector factor genes were identified (see Table A1). Common effectors include *antimicrobial peptides (AMPs)*, melanin mediated by *prophenoloxidase (PPO)*, *lysozymes (Lys)*, and *reactive oxygen species (ROS)* [35,36].

In *S. frugiperda*, five *types* of AMP genes were identified: 2 unigenes of *drosomycin*, 5 of *attacin*, 2 of *defensin*, 1 of *holotricin*, 1 of *cecropin*, and 2 of *anionic antimicrobial peptides*. Additionally, the transcriptome revealed 7 unigenes of *lysozyme*, 11 of *chitinase*, 18 of *heat shock proteins* (HSPs), and 39 of *actin*.

The results of two-way clustering of the heatmap show that immune effector factors are mainly positive-regulatory elements with high expressions in the microbial treatments. Among them, the expression levels of *Peroxidase-1, 2* and *GSH-Px-3* generally show a downward trend after infection by pathogenic microorganisms, and their high expression indicates negative-regulatory elements. In the Bt treatment, most genes show an upward trend, among which *Catalase-1, 4, 5* and *GSH-Px-4* are the most significantly upregulated. In the Sa treatment, *GSH-Px-4* and *SOD* are important positive-regulatory elements. After the Bb and Ec treatments, there are the most positive-regulatory genes, and *Peroxidase-1*, *2* and *GSH-Px-3* are important negative-regulatory elements (Figure 13).

As shown in Figure 14, *S. frugiperda* contains four *gloverin* genes, which are antimicrobial peptides specific to lepidopteran species. In the phylogenetic tree, these genes cluster with those of *H. armigera*. Most *lysozyme* genes also cluster with those of *H. armigera*, indicating a close evolutionary relationship with other lepidopteran species.

### 3.7. Analysis of the Temporal and Spatial Expression Patterns of PGRPs in S. frugiperda Induced by Microorganisms

Based on the above research, it has been shown that during the invasion of *S. frugiperda* by pathogenic microorganisms, various immune genes in *S. frugiperda* are activated to fend off infections from different types of pathogens. To further validate the accuracy of this conclusion, following the pathogen’s invasion of the insect, we selected the genes of the *PGRP*, and then analyzed their expression patterns in different tissues and developmental stages of the insect.

Experimental results indicate that in the body wall of fourth-instar *S*. *frugiperda* larvae, the expression level of the *PGRP-LE2* gene reaches its peak at 48 h after induction by Ec, which is 41 times that of the control group. In the case of Bb, the gene shows induced expression at different time points to varying degrees. In the hemolymph, the *PGRP-E2* gene is significantly activated by Ec. After 24 h and 48 h of induction, its expression levels are significantly higher than those of the control group and other treatments. At 48 h, it is significantly higher than the other treatments, with an expression level 107 times that of the control group. In the fat body, the expression of the *PGRP-LE2* gene is upregulated at all time points. After being induced by Ec, it first increases and then decreases, reaching the highest expression level at 8 h. The highest expression level after Bt induction is at 24 h, that after Sa induction is at 8 h, and that after Bb induction is also at 8 h. In the midgut, the induced expression of the *PGRP-LE2* gene is most significant only under the Ec treatment at 48 h. For the *PGRP-LB* gene, more obvious changes occur in the hemolymph after Bt induction, in the fat body after Bb induction, and in the midgut after Ec and Bb inductions. The *PGRP-LB1* gene is significantly expressed in different tissues after Bb treatment. In the body wall at 4 h, hemolymph at 4 h, fat body at 4 h, and midgut at 12 h, the expression levels are 263-fold, 67-fold, 237-fold, and 13-fold that of the control group, respectively. Under Ec induction, the expression levels in the midgut at 2 h and 8 h are significantly higher than those of the control group, while the induced expression changes of Sa in different tissues at different time points are not obvious. For the *PGRP-LB2* gene, after Bt treatment at 8 h, significant expression is observed in all tissues compared with the control group.

After treatment with Bb and Ec, the expression of *PGRP-L1* is upregulated at multiple time points. In the body wall, hemolymph, and fat body, the induction is most obvious at 24 h. In the midgut, the most significant induction by Ec is at 8 h, and the highest expression level after Bb treatment is at 48 h. After induction by different microorganisms, for the *PGRP-L2* gene, the most significant expression changes in different tissues occur at 8 h and 48 h under Ec treatment, and significant expression occurs in the midgut at 12 h under Bb treatment. The above results indicate that the *PGRP-LE2*, *LB*, *LB1*, *LB2*, *L1*, and L2 genes may be involved in the activation of the Imd pathway induced by the Gram-negative bacterium *E. coli* (Figure 15).

Twelve hours after injecting Bt, the expression levels of the *PGRP-S1* gene in the body wall and hemolymph of *S. frugiperda* larvae were higher than those of the control group and other treatment groups, being 3213 times and 2152 times that of the control group, respectively. In the midgut and fat body, the expression of *PGRP-S1* was upregulated after the four treatments, but not significantly. On the contrary, high expression was observed in the control group treated with PBS. After being induced by Bb, the expression of the *PGRP-S2* gene was significantly upregulated at multiple time points in the body wall and fat body. After being induced by Bt, significant upregulation occurred at different time points in the hemolymph, midgut, and fat body. The overall expression levels induced by Sa and Ec treatments were less obvious compared with those by Bt and Bb treatments. The expression level of the *PGRP-S3* gene was significantly higher than that of the control group and other treatments at 48 h after being induced by Sa.

Based on the above results, the expression levels of *PGRP-S1*, *S2*, and *S3* genes increased significantly after being induced by the Gram-positive bacteria *S. aureus* and *B. thuringiensis*. It is speculated that these genes may activate the Imd pathway (Figure 16).

## 4. Discussion

With the continuous advancements in next-generation sequencing, insect immunogenomics research has garnered increasing attention, expanding its focus beyond model insects such as *Drosophila* and *B. mori*. In recent years, innate immunity in insects has become a research hotspot. The components of the innate immune systems of many insects have been gradually elucidated. For example, in 2013, the immune system of the tobacco hornworm *M. sexta* was characterized, identifying 232 immune-related genes [37]. In 2014, Liu et al. identified 190 immune-related genes in the Asian corn borer *Ostrinia furnacalis* [34]. In 2015, the immune systems of the agricultural pests *Helicoverpa armigera* and *P. xylostella* were analyzed, with 233 and 149 immune genes identified, respectively [27,38]. In 2018, the immune system of the invasive pest *Dendroctonus valens* was revealed by Xu Letian’s research group at Hubei University, identifying 185 immune-related genes [39]. As a globally significant migratory agricultural pest under surveillance, the fall armyworm (*S. frugiperda*) remains poorly understood in terms of its innate immune system. However, the widespread use of *B. thuringiensis* (Bt)-based biopesticides in pest control poses a high risk of resistance development. By targeting the host immune response to pathogens and reducing its innate immunity, novel green and effective pest management strategies could be developed.

To this end, this study employed transcriptome sequencing to comprehensively analyze the composition and dynamic changes in immune genes in fall armyworm larvae under different bacterial and fungal infections. A total of 598 immune-related genes were identified, including four major categories: pattern recognition receptors, immune effectors, signal transduction factors, and immune regulators.

This study analyzed these four categories of immune factors in fall armyworms, showing that the immune gene repertoire of this species is relatively conserved, possessing a complete set of components from pathogen recognition to the production of effectors, without significant loss. This suggests that *S. frugiperda* has a well-developed innate immune defense system. However, some insects exhibit component loss during evolution, such as the absence of the JAK/STAT pathway ligand hopscotch protein in the genome of *B. mori* [23] and the absence of the Toll pathway ligand *MyD88* in the transcriptome of *P. xylostella* [38]. The systematic identification and analysis of immune gene families in *S. frugiperda* not only provide insights into their evolutionary history but also establish a theoretical foundation for further functional analysis of these immune genes.

Among pattern recognition receptors, different genes exhibited varied trends in response to pathogens, suggesting their involvement in distinct pathways for combating microbial infections. Phylogenetic analysis and expression validation revealed that *S. frugiperda PGRP-L2* clustered with *Drosophila PGRP-LD*, which is associated with maintaining gut microbial homeostasis [36]. Previous studies have shown that *BmPGRP*-S1 and *HaPGRP-A* can activate the *phenoloxidase* cascade, while *DmPGRP-LC* mediates the regulation of the *Drosophila* Imd pathway and initiates phagocytosis and Imd signaling by recognizing Gram-negative bacteria [40,41,42]. It is hypothesized that *S. frugiperda PGRP-L1* and *PGRP-S1* may share similar functions. In the transcriptome database, three *βGRP* genes were identified in *S. frugiperda*. Studies have shown that *B. mori βGRP1* can bind bacterial cell wall polysaccharides and *β-1,3-glucan*, activating *PO*-mediated melanization [43]. It is inferred that *S. frugiperda βGRP2*, which shares conserved glucan-binding domains at its N-terminus with *B. mori βGRP1*, might serve as a key receptor for bacterial detection and activation of downstream immune signaling pathways.

Immune regulators in *S. frugiperda* mainly include *serine proteases* and serine protease inhibitors, whose expression levels variy in response to pathogenic infections but function cooperatively in immune signaling. Previous studies found high similarity between *H. armigera cSP6* and *M. sexta PAP3* [25], the latter being critical for *PPO* activation. It is hypothesized that *S. frugiperda SP-59* has similar functions in melanization activation. *Serine protease inhibitors* provide feedback regulation to attenuate immune signaling. Phylogenetic analysis indicated that *S. frugiperda Serpin-3* might share similar functions with *M. sexta Serpin-6*, regulating melanization and Toll signaling pathways [27]. Signal transduction factors were mainly identified in the Toll, Imd, JAK/STAT, and JNK pathways. Toll is the primary signaling pathway in insects for defense against fungi and Gram-positive bacteria. Over evolutionary time, it has formed unique homologs, with *Spätzle* cytokines in *S. frugiperda* modulating Toll pathway activity to influence host immunity.

In the insect immune defense system, signal transduction factors play a crucial role. They are responsible for transmitting signals of external pathogen invasion into the cells to initiate an immune response. *SfurToll-1* shares the highest similarity with *DmelToll-9*, and *SfurToll-12* has the highest similarity with *DmelToll-1*. In *Drosophila*, *Toll-1* and *Toll-9* can induce the expression of antimicrobial peptides [44]. It is speculated that *SfurToll-1* and *SfurToll-12* in *S. frugiperda* may have similar functions. The *Toll-6* gene is likely to be involved in regulating the innate immune response in the midgut of *Plutella xylostella* as a member of the *Toll* signaling pathway [45]. The research results of Luna et al. indicate that the *Toll-9* gene is highly expressed in the midgut of adult *Anopheles gambiae*, and it may be involved in the innate immune response against invading pathogens [46]. *PxylToll-6* and *AgamToll-9* cluster with *SfurToll-9*, suggesting that *SfurToll-9* may be involved in the innate immune response against invading pathogens.

Toll receptors in insects do not directly detect foreign substances; instead, they function as receptors for the cytokine *Spätzle*. Studies have shown that activating *Bmorspz1* in *B. mori* increases the mRNA levels of *antimicrobial peptides* [47], and *BmorSpz4* can activate intracellular Toll signaling to enhance the host’s immunity against external infections [48]. Given the homology between *SfruSpz2* in *S. frugiperda* and *BmorSpz4* in *B. mori*, it is hypothesized that *SfruSpz2* may act as a cytokine binding to *Toll* receptors in *S. frugiperda*, regulating *Toll* pathway activity and influencing the host’s immune response.

Most immune effectors showed increased expression following pathogen infection. Phylogenetic analysis of *gloverin*, an antimicrobial peptide unique to Lepidoptera, showed that *S. frugiperda gloverin* clustered with *H. armigera*, indicating functional similarity. These adaptations likely enhance host resilience in complex and dynamic environments.

In insects, *PGRP* can recognize peptidoglycan in the bacterial cell wall and trigger the Toll and Imd pathways [49]. The peptidoglycan of Gram-positive bacteria is recognized by *PGRP*, thereby activating the Toll signaling pathway; the peptidoglycan of fungi and Gram-negative bacteria is recognized by *PGRP*, thus activating the Imd signaling pathway. The expression levels of *OfGBP* in the fat body and hemocytes of the Asian corn borer (*O. furnacalis*) are relatively high. When *Pseudomonas aeruginosa* and *Micrococcus luteus* were injected into the fourth-instar larvae of the Asian corn borer, the results of qPCR showed that the expression levels of *OfGBP* were significantly upregulated [50]. Insect *PGRPs* show expression specificity among different tissues. The *PGRP-LB* gene has the highest expression level in the midgut of adult *Bactrocera dorsalis*, followed by a relatively high expression level in the fat body of adults; the *PGRP-SB* gene is mainly expressed in the fat body of third-instar larvae and sexually mature adults [51]. *PxPGRP-S1* in *P. xylostella* has the highest expression level in the fourth instar larval stage, with the highest expression in the fat body [52]. Both *OfPGRP-A* and *OfPGRP-B* in *O. furnacalis* have the highest expression level in the midgut, which is significantly higher than that in other tissues [53]. In *Drosophila*, *PGRP-SA* can bind to Ly-type peptidoglycan and activate the Toll pathway; *PGRP-LE* and *PGRP-LC* jointly activate the IMD pathway [33,54]. In this study, among the L-type genes, the expression level of *PGRP-LE2* in tissues treated with Ec was significantly higher than that in other tissues. *PGRP-LB* and *PGRP-LB1* showed high expression levels when treated with Bb, while *PGRP-L1* and *PGRP-L2* showed high expression levels in tissues treated with Ec. Among the S-type genes, *PGRP-S1* showed high expression levels in the body wall and hemolymph when treated with Bt. *PGRP-S2* showed high expression levels in the hemolymph, fat body, and midgut when treated with Bt. The expression level of *PGRP-S3* in each tissue treated with Sa was significantly higher than that in other tissues. *S. aureus* and *B. thuringiensis* are Gram-positive bacteria containing Lys-type PGN, while *E. coli* is a Gram-negative bacterium containing DAP-type PGN. The changes in the expression levels of *PGRP* genes after being induced by different types of bacteria imply that PGRP may be involved in the regulation of the signaling pathway in some way. Combining with the previous analysis of the characteristic structures of the two genes, it is speculated that *PGRP-S1*, *S2*, *S3* may be involved in the activation of the Toll pathway, and *PGRP-LE2*, *LB*, *LB1*, *L1*, *L2* may be involved in the activation of the Imd pathway [33,55]. Through this study, the immune transcriptome of *S. frugiperda* in response to different types of pathogenic microorganisms has been basically clarified. In the follow-up, based on the differential expression changes in *PGRPs* in *S. frugiperda* after microbial infection, *PGRP* genes that have a strong response to exogenous pathogen infection and show significant expression differences will be excavated as target genes for the development of new insecticides, providing important references for improving the control efficiency of existing microbial agents and developing new, highly effective biological insecticides with specific targets [56].

## Figures and Tables

**Figure 1 insects-16-00360-f001:**
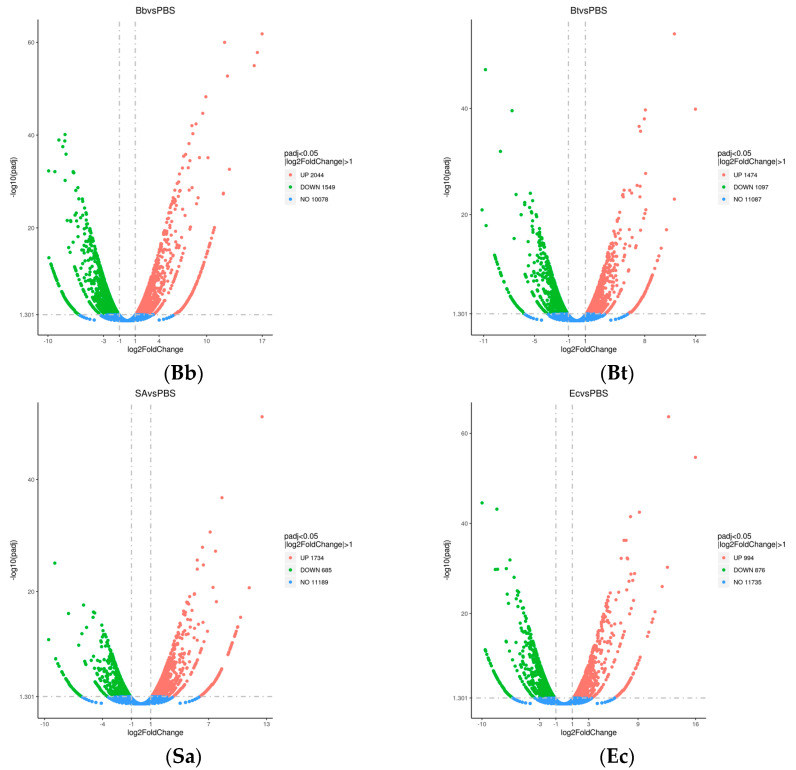
Differentially expressed gene (DEG) volcano plot of *S. frugiperda* treated with pathogenic microorganisms. Note: log2FC > 1, padj < 0.05.

**Figure 2 insects-16-00360-f002:**
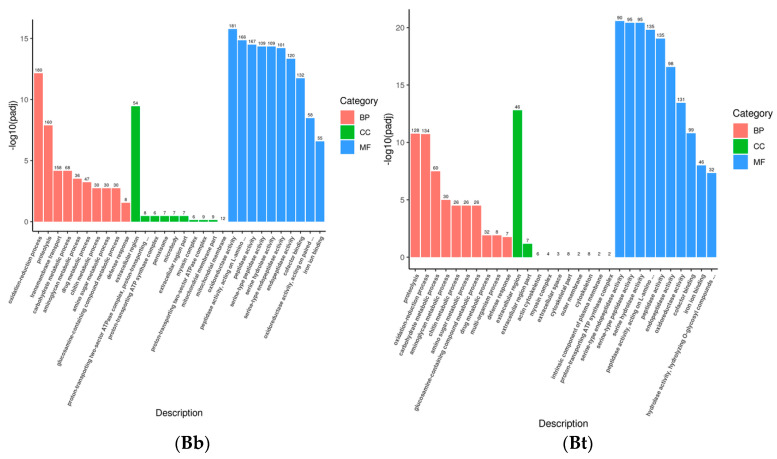
Bar charts of the GO enrichment analysis of *S. frugiperda* after being treated with microorganisms.

**Figure 3 insects-16-00360-f003:**
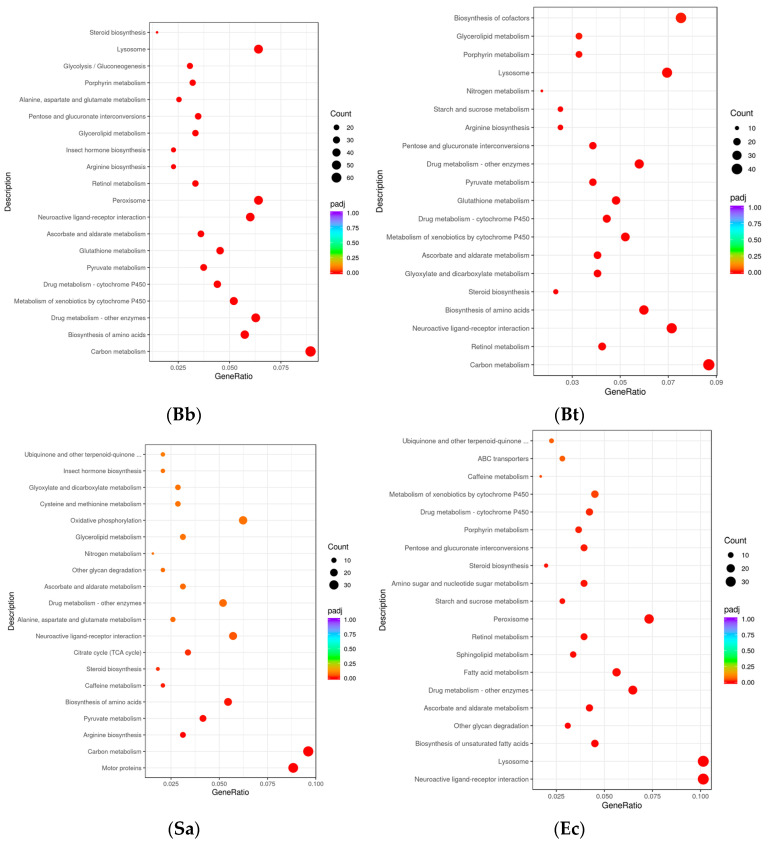
KEGG pathway enrichment analysis of differentially expressed genes in *S. frugiperda* treated with pathogenic microorganisms.

**Figure 4 insects-16-00360-f004:**
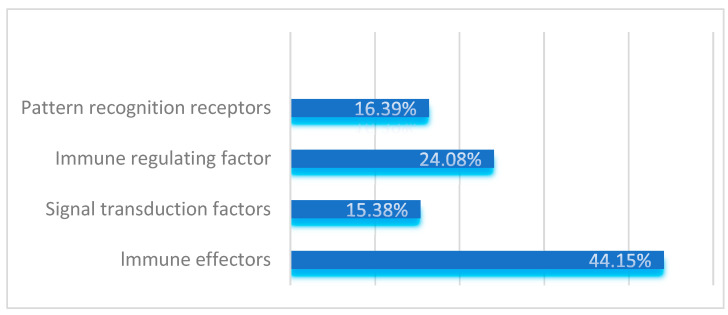
Functional distribution of immune-related genes in *S. frugiperda*.

**Figure 5 insects-16-00360-f005:**
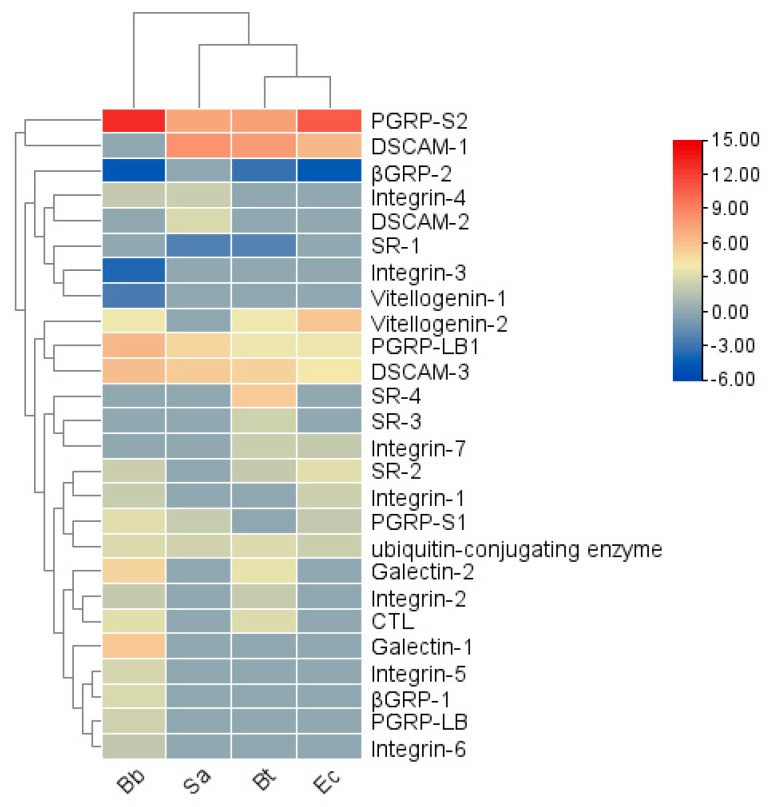
Cluster heatmap of the differential expression of pattern recognition receptors in *S. frugiperda* infected by pathogenic microorganisms. X-axis: different treatments (Bb, Sa, Bt, Ec); Y-axis: gene names. Linkage method: single linkage; clustering method: agglomerative hierarchical clustering. The numbers represent the fold changes in the genes under different treatments compared with the control.

**Figure 6 insects-16-00360-f006:**
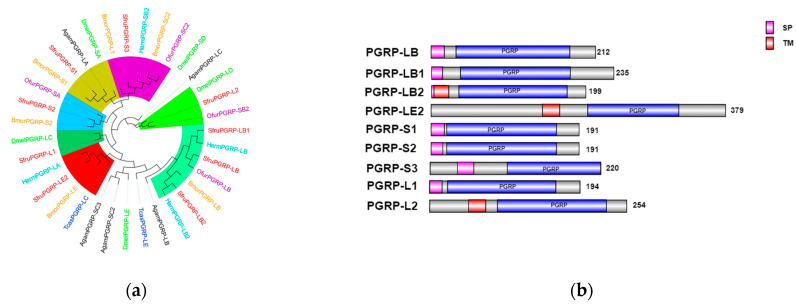
(**a**) Phylogenetic tree of *PGRP* family, in which the *PGRP* family of *S. frugiperda* is marked in red. (**b**) Sequence structure of *PGRP* protein. SP: Signal peptide, TM: transmembrane domain. Note: Insects included in the phylogenetic tree are *Aedes aegypti*; *Drosophila melanogaster*; *Bombyx mori*; *Musca domestica*; *Helicoverpa armigera*; *Ostrinia furnacalis*.

**Figure 7 insects-16-00360-f007:**
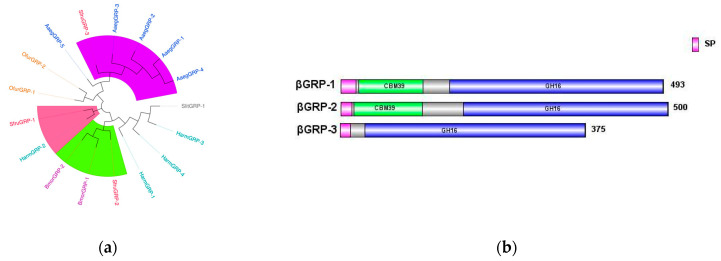
(**a**) Phylogenetic tree of *βGRP* family, in which the *βGRP* family of *S. frugiperda* is marked in red fonts. (**b**) Sequence structure of *βGRP* protein. SP: Signal peptide. Note: Insects included in the phylogenetic tree (**a**) *Aedes aegypti*; *Spodoptera litura*; *Helicoverpa armigera*; *Bombyx mori*; *Ostrinia furnacalis*.

**Figure 8 insects-16-00360-f008:**
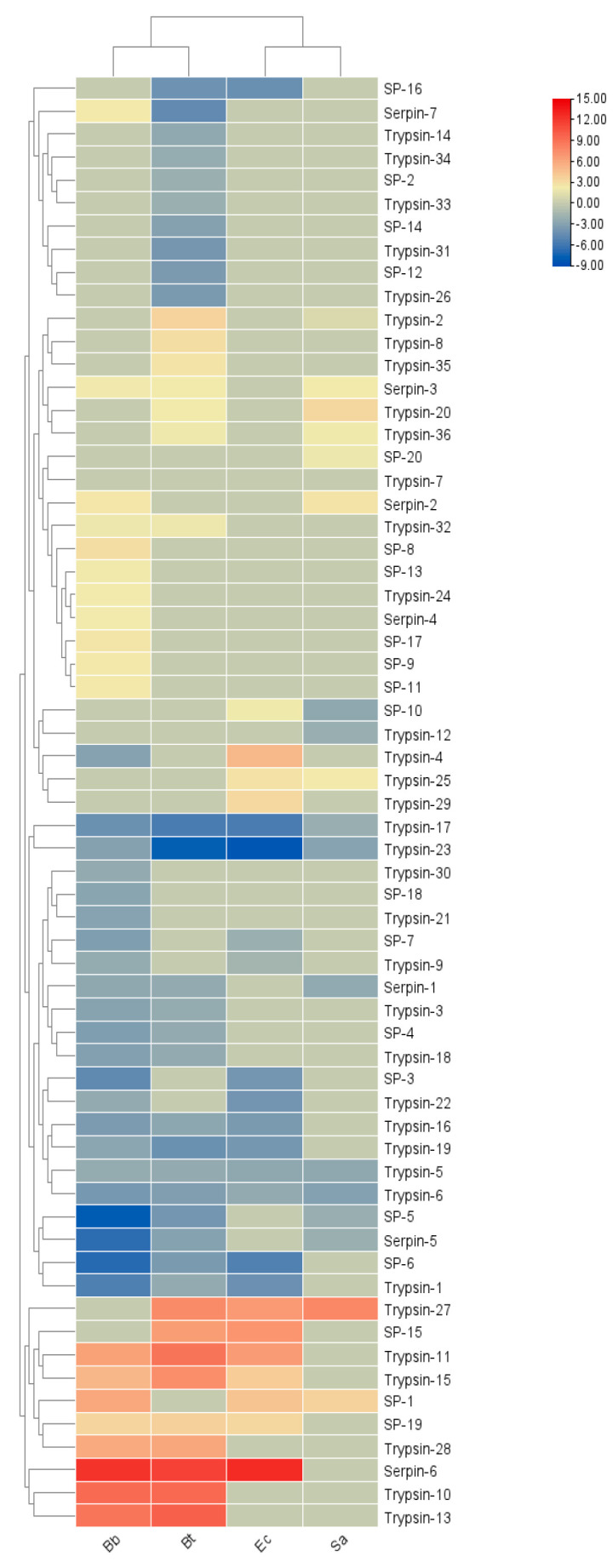
Cluster heatmap of differential expression of immunomodulatory factors in *S. frugiperda* infected by pathogenic microorganisms. X-axis: different treatments (Bb, Sa, Bt, Ec); Y-axis: gene names. Linkage method: single linkage. Clustering method: agglomerative hierarchical clustering. The numbers represent the fold changes in the genes under different treatments compared with the control.

**Figure 9 insects-16-00360-f009:**
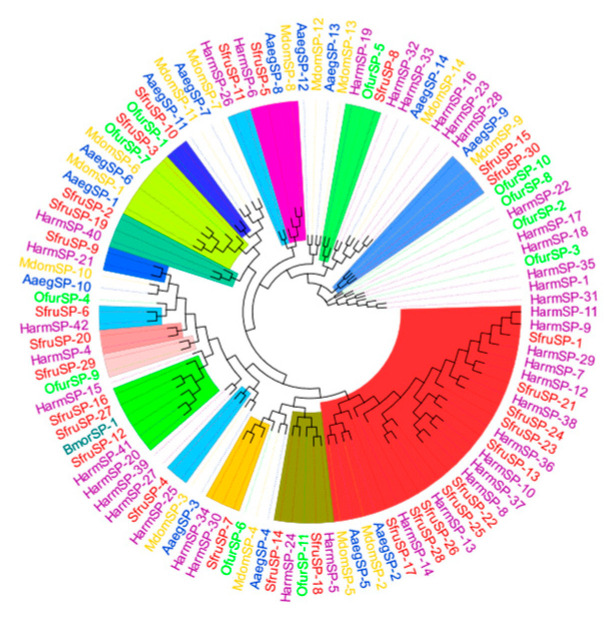
SP family phylogenetic tree, in which the SP family of *S. frugiperda* is marked with red font. Note: Insects included in the phylogenetic tree *Helicoverpa armigera*; *Bombyx mori*; *Musca domestica*; *Ostrinia furnacalis*; *Aedes aegypti*.

**Figure 10 insects-16-00360-f010:**
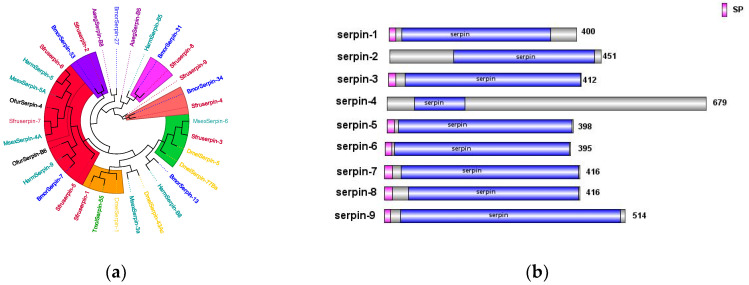
(**a**) Phylogenetic tree of the *Serpin* family, in which the Serpin family of *S. frugiperda* is marked with red fonts. (**b**) Sequence structure of the *Serpin* protein. SP: signal peptide. Note: Insects included in the phylogenetic tree: *Bombyx mori*; *Aedes aegypti*; *Drosophila melanogaster*; *Ostrinia furnacalis*; *Helicoverpa armigera*; *Manduca sexta*; *Tenebrio molitor*.

**Figure 11 insects-16-00360-f011:**
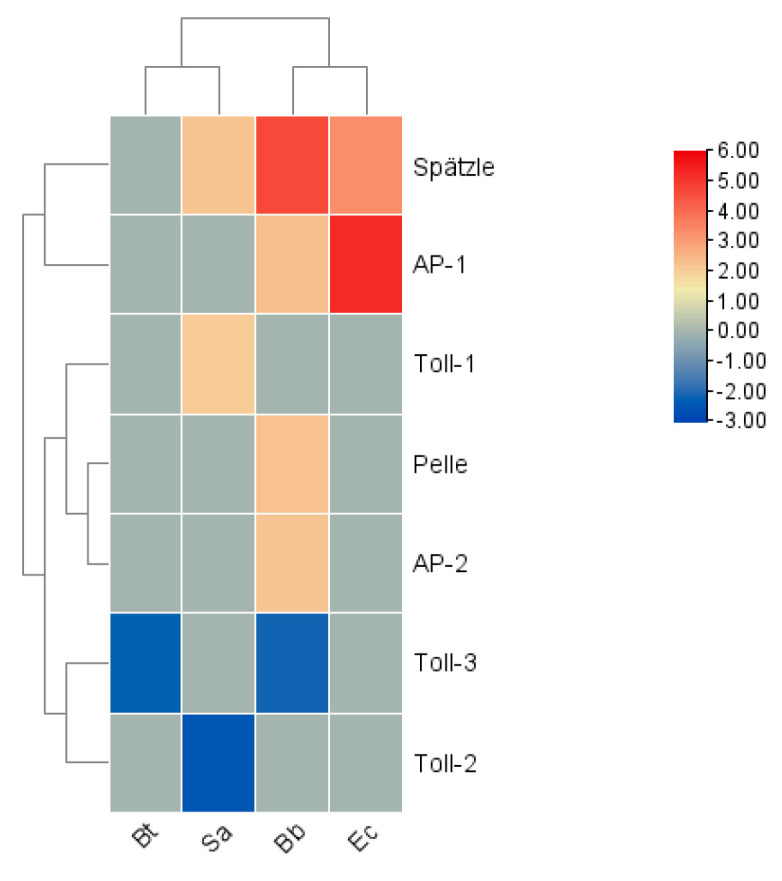
Cluster heatmap of the differential expression of signal transduction factors in *S. frugiperda* infected by pathogenic microorganisms. X-axis: different treatments (Bb, Sa, Bt, Ec); Y-axis: gene names. Linkage method: single linkage; clustering method: agglomerative hierarchical clustering. The numbers represent the fold changes in the genes under different treatments compared with the control.

**Figure 12 insects-16-00360-f012:**
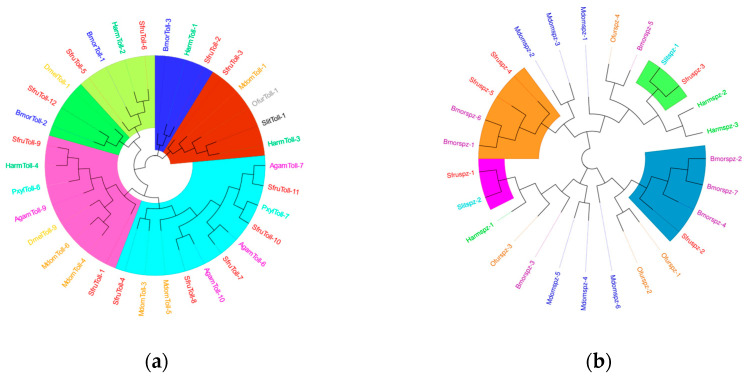
(**a**) The phylogenetic tree of the *Toll* family. (**b**) The phylogenetic tree of the *Spz* family, in which the *S. frugiperda*-related family is marked with red font. Note: Insects included in the phylogenetic tree: (**a**) *Musca domestica*; *Ostrinia furnacalis*; *Spodoptera litura*; *Helicoverpa armigera*; *Bombyx mori*; *Aedes aegypti*; *Drosophila melanogaster*; *Plutella xylostella*; (**b**) *Musca domestica*; *Spodoptera litura*; *Helicoverpa armigera*; *Bombyx mori*; *Ostrinia furnacalis*.

**Figure 13 insects-16-00360-f013:**
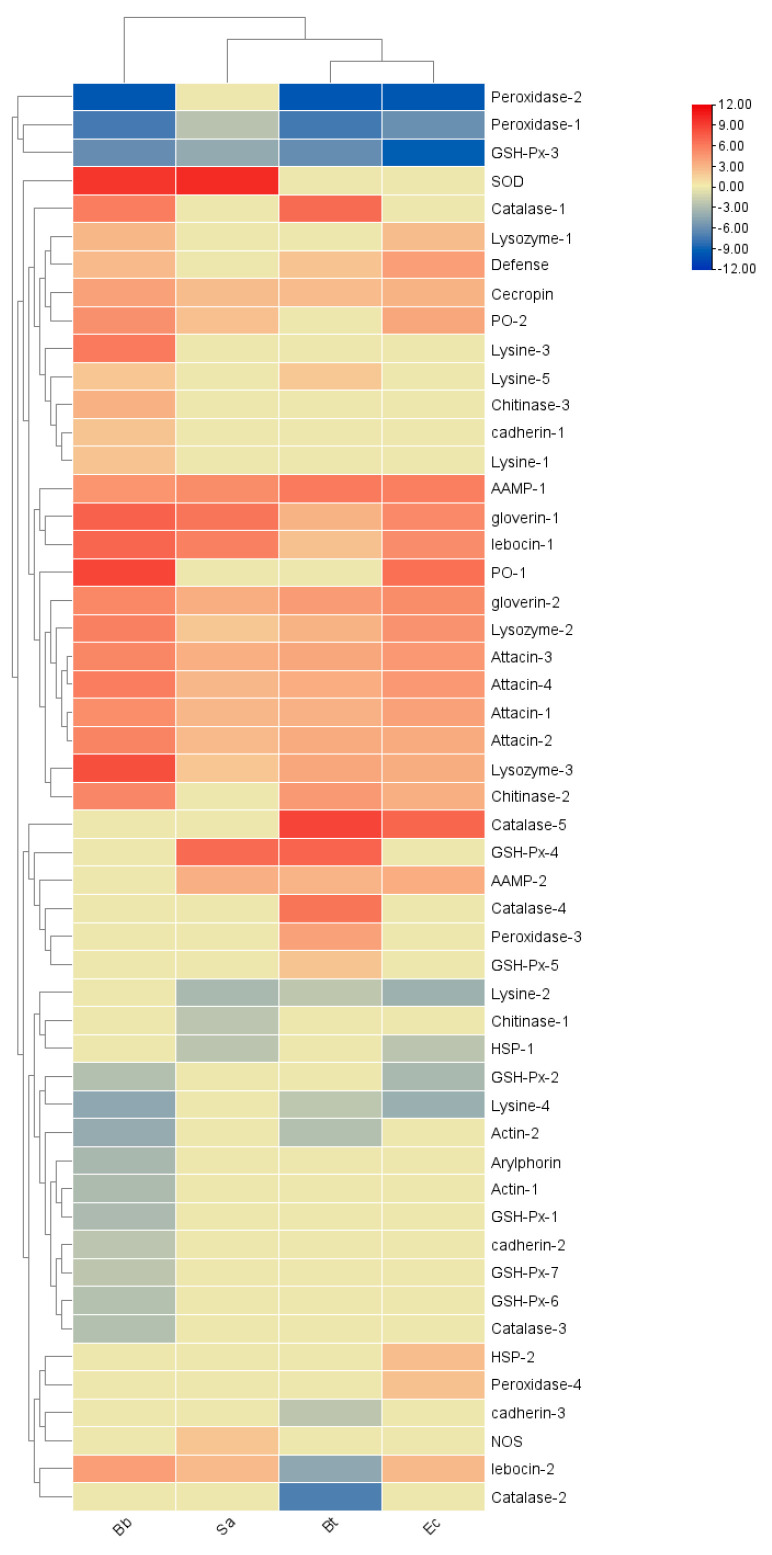
Cluster heatmap of differential expression of immune effectors in *S. frugiperda* infected by pathogenic microorganisms. X-axis: different treatments (Bb, Sa, Bt, Ec); Y-axis: gene names. Linkage method: single linkage; clustering method: agglomerative hierarchical clustering. Whether to perform row/column clustering: Yes, to display the similarities between rows and between columns. The numbers represent the fold changes in the genes under different treatments compared with the control.

**Figure 14 insects-16-00360-f014:**
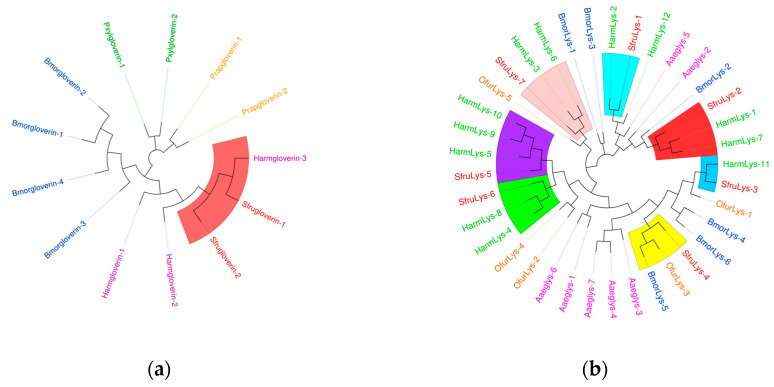
(**a**) The phylogenetic tree of the gloverin family. (**b**) The phylogenetic tree of the lysozyme family, in which the *S. frugiperda* related family is marked with red fonts. Note: (**a**) *Bombyx mori*; *Plutella xylostella*; *Pieris rapae.* (**b**) *Helicoverpa armigera*; *Ostrinia furnacalis*; *Bombyx mori*; *Aedes aegypti*.

**Figure 15 insects-16-00360-f015:**
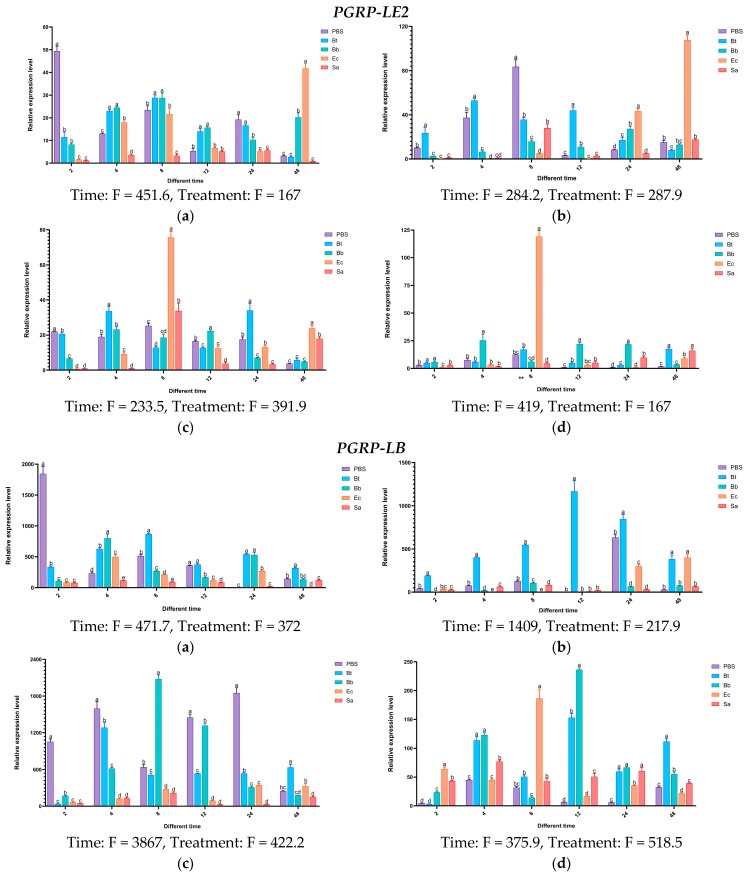
Tissue expression profiles of *PGRP-L* type in *S. frugiperda* at different time points after induction by pathogenic microorganisms. Notes: The error bars refer to the standard error. (**a**) Body wall: The expression status of the body wall at different times after treatment with pathogenic microorganisms; (**b**) hemolymph: The expression status of hemolymph at different times after treatment with pathogenic microorganisms; (**c**) fat body: The expression status of the fat body at different times after treatment with pathogenic microorganisms; (**d**) midgut: The expression status of the midgut at different times after treatment with pathogenic microorganisms. PBS: enzyme-free and sterile water; Bt: *Bacillus thuringiensis*; Ec: *Escherichia coli*; Sa: *Staphylococcus aureus*; Bb: *Beauveria bassiana*. Note: Statistical test results of the two-way ANOVA for the interaction between time and treatment: Time: dF = 4, *n* = 6, *p* < 0.01; Treatment: dF = 5, *n* = 5, *p* < 0.01. See the F values in the figure.

**Figure 16 insects-16-00360-f016:**
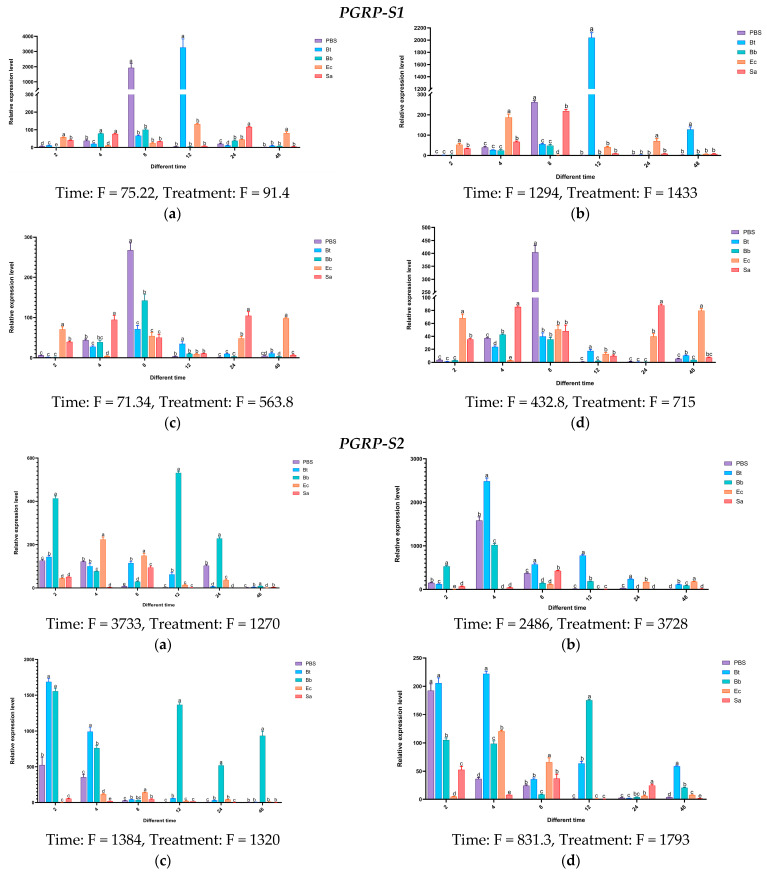
Tissue expression profiles of *PGRP-S* type in *S. frugiperda* at different time points after induction by pathogenic microorganisms. Notes: The error bars refer to the standard error. (**a**) Body wall: The expression status of the body wall at different times after treatment with pathogenic microorganisms; (**b**) hemolymph: The expression status of hemolymph at different times after treatment with pathogenic microorganisms; (**c**) fat body: The expression status of the fat body at different times after treatment with pathogenic microorganisms; (**d**) midgut: The expression status of the midgut at different times after treatment with pathogenic microorganisms. PBS: enzyme-free and sterile water; Bt: *Bacillus thuringiensis*; Ec: *Escherichia coli*; Sa: *Staphylococcus aureus*; Bb: *Beauveria bassiana*. Note: Statistical test results of the two-way ANOVA for the interaction between time and treatment: Time: dF = 4, *n* = 6, *p* < 0.01; Treatment: dF = 5, *n* = 5, *p* < 0.01. See the F values in the figure.

**Table 1 insects-16-00360-t001:** qRT-PCR primer information.

Primer	Forward Primer	Reverse Primer	Amplification Efficiency
*PGRP-LE2*	*ATTTCGCACACTGCTACCGA*	*TGGACTGAGAGTAGACGCCA*	98.34%
*PGRP-LB*	*CAAGGAAGACTGCTCAGCGA*	*AGGCAGTTCCAGGACATTCG*	95.67%
*PGRP-LB1*	*GCACGCGCTACATTTCAACA*	*TTGAAGAG-TGCGTCTCCTGG*	99.35%
*PGRP-LB2*	*AGACCGCCTAATGGTTCGAC*	*AGCCAAGCTTCACTCCAGTC*	104.63%
*PGRP-L1*	*AGCAGCCAATGGAATCAGGA*	*GAGAGCTGACTATGGGCCAC*	99.72%
*PGRP-L2*	*GTCAGCTTGCTCCTGGTGAT*	*ATCGTTCCGTTCCCGTTTGA*	96.64%
*PGRP-S1*	*AAATGGGGACTGTGGCGTAG*	*CGTATACTTTGCCGTTGCCG*	99.61%
*PGRP-S2*	*TTGTGTCGAGGATCGGTTGG*	*CTCATACACTGTCCCCTGGC*	96.38%
*PGRP-S3*	*GAATTGCGCAGCTGAGATGG*	*CAAGCTCGACACCCTTGTCT*	102.36%
*PRL18*	*GCCAAGACCGTTCTGCTGC*	*CGCTCGTGTCCCTTAGTGC*	108.78%
*PRL3*	*CCAAGGGTAAAGGATACAAAGGTG*	*TCATTCACCGTTGCCCGT*	97.93%

**Table 2 insects-16-00360-t002:** Evaluation statistics of the transcriptome data of *S. frugiperda* larvae treated with pathogenic microorganisms.

Sample	Raw Reads	Raw Bases	Clean Reads	Clean Bases	Error Rate (%)	Q20 (%)	Q30 (%)	GC Pct (%)
PBS	54,529,714	8.18 Gb	50,261,762	7.54 Gb	0.01	98.49	95.76	45.95
Ec	48,893,730	7.33 Gb	46,235,856	6.94 Gb	0.01	98.67	96.11	46.7
Bt	48,788,450	7.32 Gb	45,953,424	6.89 Gb	0.01	98.46	95.72	45.24
Sa	51,879,208	7.78 Gb	49,723,532	7.46 Gb	0.01	98.67	96.2	45.3
Bb	46,291,118	6.94 Gb	44,794,776	6.72 Gb	0.01	98.56	95.96	45.15

**Table 3 insects-16-00360-t003:** Statistics of sample alignment with the reference genome.

Sample	Total Reads	Total Map	Unique Map	Multi Map	Positive Map	Negative Map	Splice Map	Unsplice Map
PBS	50,261,762	40,753,040 (81.08%)	37,813,038 (75.23%)	2,940,002 (5.85%)	18,879,122 (37.56%)	18,933,916 (37.67%)	15,439,827 (30.72%)	22,373,211 (44.51%)
Ec	46,235,856	38,147,549 (82.51%)	35,977,282 (77.81%)	2,170,267 (4.69%)	17,976,673 (38.88%)	18,000,609 (38.93%)	14,916,240 (32.26%)	21,061,042 (45.55%)
Bt	45,953,424	36,771,734 (80.02%)	34,942,889 (76.04%)	1,828,845 (3.98%)	17,461,099 (38.0%)	17,481,790 (38.04%)	13,877,738 (30.2%)	21,065,151 (45.84%)
Sa	49,723,532	39,520,310 (79.48%)	37,315,555 (75.05%)	2,204,755 (4.43%)	18,638,812 (37.48%)	18,676,743 (37.56%)	14,594,876 (29.35%)	22,720,679 (45.69%)
Bb	44,794,776	35,009,774 (78.16%)	33,124,498 (73.95%)	1,885,276 (4.21%)	16,548,807 (36.94%)	16,575,691 (37.0%)	13,067,821 (29.17%)	20,056,677 (44.77%)

## Data Availability

The data presented in this study are available on reasonable request from the corresponding author.

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
