# Peer review of "Immunotranscriptomic Profiling of Spodoptera frugiperda Challenged by Different Pathogenic Microorganisms"

_insects, 2025, doi:10.3390/insects16040360_

Round 1
Reviewer 1 Report (Previous Reviewer 1)
Comments and Suggestions for Authors
The authors have done a fine job of refining their manuscript, improving clarity, explanations of methods, and general scientific accuracy.
Normalization of gene expression relies on FPKM, which is outdated and does not account for sequencing depth and variability well; TPM or DESeq2 normalization would be more accurate.
The argument would be enhanced by greater detail regarding how the genes found to be implicated in the immune response make functional contributions to host-pathogen interactions, complemented by comparative data from close relatives.
-------------
The RNA extraction process should explicitly state if rRNA depletion or poly(A) selection was used.
Statistical validation of differential expression analysis does not actually mention multiple comparison corrections (e.g., FDR correction), leaving potential false positives behind.
Figures, particularly heatmaps, needed to have more distinct axis labels and clustering parameters for better interpretability.
Gene ontology and KEGG pathway enrichment information requires more clarification on the relation to immune responses.
Availability of raw sequencing data and metadata for reproducibility is not defined if they are deposited in a public database or not.
Part of the references is outdated and needs to be replaced with more recent literature to make the context up-to-date.
Author Response
Please see the attachment.

Reviewer 2 Report (Previous Reviewer 2)
Comments and Suggestions for Authors
This study systematically investigated the immune transcriptome characteristics of Spodoptera frugiperda larvae under bacterial (Staphylococcus aureus, Bacillus thuringiensis, Escherichia coli) and fungal (Beauveria bassiana) infections using transcriptomic sequencing and molecular biology techniques.
Line 153–155 (Section 2.5):Vague database description ("IKB database"). Missing version, access date, and search parameters.
​Line 519–525 (Section 3.6.4):Lack of functional validation for identified antimicrobial peptides (AMPs). Listing quantities (e.g., "53 putative AMPs") without activity assays weakens mechanistic claims.
​Line 240 (Figure 1):Volcano plot lacks clear significance thresholds (e.g., log2FC=1, FDR=0.05), making differential gene selection criteria ambiguous.
​Line 339 (Gene nomenclature): Non-italicized gene symbols (e.g., "SfurSP-5"). Missing NCBI accession numbers.
​Line 233 (Table 2): Incorrect unit spacing ("7.54G" instead of "7.54 Gb").
Author Response
Please see the attachment.

Reviewer 3 Report (New Reviewer)
Comments and Suggestions for Authors
The manuscript describe immunotranscriptomic response of S. frugiperda challenged with different pathogens. The data are interesting but they should be shared with scientific community. However authors did not provide any link to repository nor to NCBI reads archive. Moreover the manuscript should be revised in many part, by providing detailed explanation of some methodologies used and a clearer description of results. Several parts, such as the final results paragraph, need to be re-organized (figures) and rewritten (main text). I'm attaching a pdf version of the manuscript which includes several suggestions as comments.

Comments on the Quality of English Language
English is ok, but can be improved.
Author Response
Please see the attachment.

This manuscript is a resubmission of an earlier submission. The following is a list of the peer review reports and author responses from that submission.
Round 1
Reviewer 1 Report
Comments and Suggestions for Authors
Tang et al. profiled the immune transcriptome of Spodoptera frugiperda against microbial infection and identified 655 immune-related genes, which further classified into functional groups. Although this study has succeeded in using transcriptomics to uncover some immune-related pathways, it lacks experimental robustness, interpretative depth, and clarity. There were methodological issues, inadequate validation, and overstated conclusions that diminished the reliability and translational relevance of the findings. The introduction is well justified, but it does not take into consideration the insights of previous studies. The discussion repeats the results without a proper critique of limitations and integration with the wider literature. Language quality is clear and well-flowing; 8/10. Overall, this work deserves 72/100 for its contribution to insect immunity, but serious methodological and interpretative deficiencies seriously limit the reliability and applicability of the research findings. Refocusing the study should include tissue-specific analyses, functional assays, and realistic pathogen exposures to enhance biological relevance and practical utility.
__________________________________________
Major concerns
1. Methods for microbial preparation and injection protocols are not well-defined and, in fact, are contradictory. The text switches between "inactivated bacterial suspension" on line 106 and live bacterial cultures prepared to OD600 ~1.0, leaving the question as to whether immune activation was due to live pathogens or dead microbial components. This biological irrelevance of the data thus interprets the immune gene responses.
2. Whole-body transcriptomics dilutes tissue-specific immune signals, and, therefore, precise conclusions on the dynamics of immune responses are not possible. For example, AMPs, which are mainly produced in hemocytes and fat bodies, may appear underrepresented due to the contribution of non-immune tissues such as gut and muscle. This is a specific problem for claims about upregulated immune genes across the treatment groups, for instance, line 227.
3. The 655 reported immune-related genes are not sufficiently validated for annotation. For instance, the identification of 20 Toll pathway genes (line 360) includes phylogenetic clustering inconsistencies in Figure 12a, which suggests redundancy likely stemming from reference genome duplication errors. Containing 12.7% duplication, the likely genome source, Xiao et al. (2020), risks inflating immune repertoire counts and making spurious gene discoveries.
4. KEGG pathway enrichment analysis identifies pathways such as "lysosome" and "neuroactive ligand-receptor interaction" (lines 197-204) but does not discuss their specific contributions to immune responses. For instance, lysosomes play a role in AMP processing or autophagy in the context of infection resolution, which does not relate to any such processes identified from gene expression profiles, hence limiting the biological relevance of these findings.
5. DEG is described using thresholds of FDR < 0.01 and fold change >2 (lines 126-127), yet figures like Figure 11 contain genes with very small fold changes, indicating a lax application of statistical criteria. Besides, the qPCR fold differences are more than 500-fold in certain cases, like that of PGRP-S1 under Bt treatment, line 438, sharply diverging from the trends of RNA-seq and raising questions about the reliability of normalization.
6. Claims regarding Toll and Imd pathway activation (lines 346-349) are still speculative because no functional validation was provided. The study identifies several key regulators, such as PGRP-S1 and βGRP-1, but fails to perform knockdown or overexpression studies, let alone protein-level studies to validate their roles. This makes the conclusions on immune signaling incomplete and hypothetical.
7. The microbial treatments lack pathogen-specific analysis. Although there is differential immune gene expression between bacterial (Bt, E. coli, S. aureus) and fungal (B. bassiana) infections, the heatmaps, for example, Figure 5, obscure pathogen-specific immune dynamics by conflating patterns across pathogens without highlighting gene subsets specifically linked to bacterial or fungal recognition.
8. The focus of the authors on fourth-instar larvae is not justified. Although this is a standard stage in laboratory studies, immunity can vary between different stages, especially between larvae, pupae, and adults. The results are not generalizable since developmental immunity, one of the most crucial variables, has not been considered or checked.
9. Phylogenetic analyses overemphasize evolutionary conservation without addressing functional divergence. For example, βGRP genes cluster with those from Bombyx mori and Helicoverpa armigera, lines 287-301, without further experimental validation of functional similarities. This gap avoids valid conclusions on the evolution of immune genes and their function.
10. One of the biggest limitations is the lack of raw RNA-seq data availability in public repositories. Accession numbers or links to the datasets are not provided, which will prevent the reproducibility and independent validation of findings (lines 552-553).
11. The pipeline for identifying genes does not provide details on reproducibility. For example, although the manuscript states that immune-related genes were identified from previous descriptions (line 160), it does not explain the criteria, reference studies, or steps for manual curation. The absence of such information renders the immune repertoire claims unverifiable.
12. Redundancy within the immune gene repertoire is not addressed. For example, it does not distinguish between the immune-specific and other physiological functions of the 61 serine proteases identified (line 314), which may overestimate the complexity of S. frugiperda immunity.
13. The sampling protocols preclude any possibility of immune response variability at the individual level. The approaches pool larvae (line 101), thus removing individuality and probably distorting the expression profiles for genes responding, especially in immune-challenge experiments. These will likely introduce potential sources of bias in trends based on the observations.
14. The identified immune genes to be used as targets of biopesticides-since lines 536-540 do not show experimental validation or field-level feasibility studies. For example, PGRP-S1 and βGRP-2 are suggested as targets; however, their functional involvement in survival or resistance against pathogens is not tested directly. Hence, such proposals are premature and speculative.
15. Temporal dynamics are not included in the design of the experiment, thus allowing only limited interpretations of the patterns of immune gene activation. The study tests the expression of genes at only one time point, 24 h post-infection, a time that can miss both the early and the late crucial phases of immune activation (line 110). Immune responses are typically biphasic or time-course dependent, where certain genes reach peak expressions within hours, others sustain or decrease over days, and vice versa. For instance, Toll and Imd pathways commonly show the pattern of rapid activation followed by their regulatory dampening, which clearly cannot be reflected here. Without any kind of time-course analysis, the conclusion related to immune pathway involvement and their effector roles may be incomplete, thus misinforming results. This oversight critically reduces the capacity of this study in providing a holistic view on immunity in S. frugiperda.
__________________________________________
Minor concerns:
1. While the qPCR section refers to the primer sequences in Table 1, there are no details regarding primer efficiency tests or any validation on their specificity to confirm the correctness of amplification.
2. There is an agarose gel running of RNA quality, with no reporting on RNA Integrity Number scores being done; this is a pretty standard measure for RNA quality, at least in most transcriptomics studies today (lines 113-115).
3. Figure 4 combines all immune-related gene categories into a single chart and does not show the relative abundance or differential expression patterns between treatments, which makes it less clear to explain the data presented.
4. Figures (e.g., 15 and 16) use symbols such as “*” and “ns” for statistical significance but do not define thresholds (e.g., FDR or p-values), making the interpretation of the results ambiguous.
5. While the methods indicate three replicates per treatment group (line 103), it does not mention whether these represent true biological replicates or pooled samples, which is of the utmost importance to evaluate reliability and reproducibility.
6. Figures 5 and 11 depict heatmaps of color gradients that are different; it confuses the comparisons across figures in gene expression patterns, leading to a decrease in coherence regarding data presentation.
7. Though microbial treatments were standardized to 1×10^6 cells/mL, lines 94-95, no indication is given of post-preparation assessment of microbial viability; this could lead to variable immune activation and confounding of results.
8. The paper has avoided citing Gouin et al. (2017), which annotated S. frugiperda immune genes and provided a very important base for similar studies. Limiting the depth of context in the manuscript.
9. It identifies sixty-one serine protease genes (line 314) with no discussion to enlighten on whether they are immune-specific or rather structural proteins that are misclassified due to problems with the reference genome, inflating the gene counts.
10. The references are not properly formatted, as are many others, for instance [10] and [18]; this detracts from the professionalism of the manuscript and complicates any verification of the claims presented.
11. The methods lack statistical approaches regarding GO and KEGG enrichment analyses, such as the hypergeometric test and FDR corrections. This will raise questions about reproducibility and the reliability of pathway-level conclusions (line 129).
12. Bacterial strains are mentioned, such as Staphylococcus aureus and E. coli, and fungal strains like Beauveria bassiana, but exact subspecies or isolate numbers are not mentioned, which complicates experimental reproducibility and limits the generalizability of the findings (line 89).
13. The sequencing depth and read count per sample, which are of key importance to estimate the accuracy and comprehensiveness of the transcriptomic analyses performed in this study, are not shown in the manuscript (lines 113–117).
14. Technical terms such as "pattern recognition receptors" and "immune pathways" were used; it makes no sense for an ordinary man reading "Simple Summary" lines 12–22.
15. Figures that have supplementary data, such as Appendix 1, providing a listing of immune-related genes, are not referenced through a table or figure number within the body of the work, thereby ensuring ease of information cross-reference for the reader.
16. AMPs have been claimed as being "highly expressed" between lines 399 to 401. No statistical comparisons among the treatments in proof of this were included.
17. The assertion of "completeness" for the Toll pathway, as mentioned on lines 353–354, is a claim that cannot be confirmed in the absence of functional assays or detailed analyses of nonrepresented or nonfunctional components. This assertion is, thus, speculative.
18. The overall completeness and quality of the dataset are called into question since no RNA-seq mapping metrics are provided, for instance, the proportion of the reads aligned to the transcriptome or genome, lines 120–123.
19. Figures 5 and 12 are of low resolution; obscured gene labels and in-depth analysis make important data visualizations difficult to see and understand.
20. The authors provide the implication that PGRPs are targeted genes for bio-pesticides but fail to discuss the practical matters such as off-target effects or difficult field validations that would make this an actuality (lines 536–540).
__________________________________________
While this study provides insights into S. frugiperda immunity by cataloging immune-related genes (intersect with other available literature in this respect), the experimental and methodological deficiencies noted here call for a radical refocus and substantial revisions. I recommend refocusing the work to fill in these critical gaps, increase transparency, and enhance the rigor and applicability of the study.
Upon refocusing, the work can be improved by standardization of microbial preparation procedures for making the data underpinning more biologically relevant, by tissue-specific transcriptomics to remove immune-gene dynamics ambiguities, or by rigorous immune-gene repertoire validation with special care given to phylogenetic discrepancies. Time-course experiments and functional validation go a long way toward helping support the claims made of activation of immune pathways. Public deposition of RNA-seq data is necessary so as to ensure reproducibility and transparency. These changes would make the manuscript much stronger, more transparent, and potentially more impactful.
Reviewer 2 Report
Comments and Suggestions for Authors
This study focuses on transcriptome sequencing of S. frugiperda infected by different pathogenic microorganisms, identifying immune-related genes through differential expression analysis, and conducting bioinformatic analyses and expression validation of these genes. This study elucidates the immune transcriptome of S. frugiperda in response to various pathogenic microorganisms, providing valuable insights for improving the effectiveness of existing microbial agents and developing new highly efficient and specific biopesticides. The approach of the authors is well executed, the article is well-written and the applied methodology merits publication in Biotechnology Journal. However, after reading the manuscript I have some minor and major comments (mentioned below) that ought to be addressed before publication.
1. Line 33-34: “PGRP””βGRP””qPCR” the full name should be written when it appears for the first time.
2. LIne 53-55:This sentence is not fluent, please rewrite it.
3. Line 84:Please write down the composition and proportion of artificial feed.
4. Line 85:Please explain clearly a photoperiod of 14L:10D.
5. Line 87-89:Staphylococcus aureus (SA) and B. thuringiensis (Bt), Gram-negative bacteria E. coli (Ec) were purchased from the Institute of Microbiology, Chinese Academy of Sciences. Please indicate their specific strain numbers.
6. Line 91:OD600 should be “OD600”.
7. Line 93: “Na2HPO4” and “NaH2PO4” should be changed to “Na2HPO4” and “NaH2PO4”.
8. LIne 116-117:The instrument should be marked with model and company name, etc.
9. Figure 15, Figure 16 shows that the significant difference is wrong. The difference is the statistical analysis between two or more groups of samples. The asterisk should not be marked on a set of data.
10. LIne 490-491:The first time an abbreviation appears, its full name must be indicated.
